# Biodegradable Molybdenum (Mo) and Tungsten (W) Devices: One Step Closer towards Fully-Transient Biomedical Implants

**DOI:** 10.3390/s22083062

**Published:** 2022-04-15

**Authors:** Catarina Fernandes, Irene Taurino

**Affiliations:** 1Micro and Nano-Systems (MNS), Department of Electrical Engineering (Micro- and Nano Systems), Katholieke Universiteit Leuven (KU Leuven), 3000 Leuven, Belgium; irene.taurino@kuleuven.be; 2Semiconductor Physics, Department of Physics and Astronomy (Semiconductor Physics), Katholieke Universiteit Leuven (KU Leuven), 3000 Leuven, Belgium

**Keywords:** transient electronics, fully-transient biomedical systems, tungsten, molybdenum, electrochemical sensors, remote health monitoring

## Abstract

Close monitoring of vital physiological parameters is often key in following the evolution of certain medical conditions (e.g., diabetes, infections, post-operative status or post-traumatic injury). The allocation of trained medical staff and specialized equipment is, therefore, necessary and often translates into a clinical and economic burden on modern healthcare systems. As a growing field, transient electronics may establish fully bioresorbable medical devices capable of remote real-time monitoring of therapeutically relevant parameters. These devices could alert remote medical personnel in case of any anomaly and fully disintegrate in the body without a trace. Unfortunately, the need for a multitude of biodegradable electronic components (power supplies, wires, circuitry) in addition to the electrochemical biosensing interface has halted the arrival of fully bioresorbable electronically active medical devices. In recent years molybdenum (Mo) and tungsten (W) have drawn increasing attention as promising candidates for the fabrication of both energy-powered active (e.g., transistors and integrated circuits) and passive (e.g., resistors and capacitors) biodegradable electronic components. In this review, we discuss the latest Mo and W-based dissolvable devices for potential biomedical applications and how these soluble metals could pave the way towards next-generation fully transient implantable electronic systems.

## 1. Introduction

Electronic devices have proven to be indispensable in today’s modern society [1]. The increasing demand for miniaturized, enhanced, and user-friendly devices has instigated the rapid evolution of technology we have witnessed in the past decades. Transient electronics is an emerging technology boosted by the demand for “zero-waste” consumable electronics [2]. The field of transient electronics has, since 2012 [3], launched important guidelines towards establishing unique biodegradable devices, characterized by their ability to physically dissolve without a trace in an array of environments [1]. Although long-time operation and physical invariability is a hallmark of traditional electronic devices and integrated circuits [3], transient devices could have a significant impact in the environmental sector, consumer electronics, and healthcare.

Increasing interest from both industry and academia has led to a growing body of comprehensive studies on transient materials, a natural requirement in physically soluble systems. Within this class of materials, transition metals molybdenum (Mo) and tungsten (W) have been recently classified as promising candidates to take into consideration for the fabrication of transient electronic devices. Several published studies have attested to the safe and innocuous bio-dissolution of Mo and W into non-toxic by-products, capable of biological extraction [4,5]. Moreover, Mo and W exhibit predictable and controllable dissolution mechanisms in biofluids, at physiological temperature, as well as more gradual dissolution kinetics in comparison to magnesium (Mg) or zinc (Zn) [4]. Although highly biocompatible and of value in a variety of structural implants, Mg and Zn dissolve very quickly in physiological environments (≈0.5–3 μm h^−1^), which gives rise to fleeting device lifetimes, and unstable functionality in vivo [4]. Mo and W are extremely promising metals for the fabrication of implantable, fully biodegradable, electronic systems with appropriate device lifetimes (days-weeks) and stable signal outputs during physiological dissolution, devoid of adverse immune responses [6]. Nowadays, a few examples of biodegradable Mo and W-based miniaturized electronic components can be found in the literature. Collectively, these components (e.g., power sources, interconnects, electrodes, and wires) are compulsory for the proper functionality, stable operation, and self-sustainment of electronically-active devices, and naturally, of active implants.

Despite recent advances, however, there is a clear shortage of Mo and W-based electrochemical sensors. The sensing interface is a central piece in any implantable system, which aims to locally detect physiological analytes, and transmit the electrical signals to the exterior: providing vital diagnostic and therapeutic information, either in a clinical setting or remotely. Thus, the lack of implantable fully-transient electronic systems, functioning as diagnostic or therapeutic tools, is more noticeable still. The need for a multitude of biocompatible and biodegradable electronic components, in addition to the electrochemical biosensing interface, has halted the arrival of these systems i.e., systems capable of sustaining the physiological dissolution of all their components, passive and active. In actuality, the majority of published biodegradable electronic implants with sensing capabilities can only uphold the physiological dissolution of the integrated sensor, which intrinsically must be in contact with biological tissue, leaving the rest of the components forming the system (such as transcutaneous wires and circuitry) to be surgically extracted afterwards. Overall, the pairing of fully-biodegradable system sub-components: sensor and supporting electronics (as seen in Figure 1) is compulsory to realize an entirely biodegradable functional biomedical tool, with extended lifetimes. This can be achieved by steadily incorporating existing Mo and W-based electronic components, thought of as small pieces of a larger puzzle, together with a likewise Mo and W-based electrochemical sensing interface.

In sum, this review provides an overview of existing Mo and W-based biodegradable components, and discusses how collectively, these might integrate in fully transient electronically active implantable systems. First, we will discuss the value of device transiency in a biomedical context. Second, important characteristics (e.g., physiological dissolution profiles, resistance to corrosion, biocompatibility, and bioresorbability) that dictate the usage of Mo and W in these systems are reviewed and balanced against other recurrently used biodegradable metals. Third, we describe relevant examples of recently achieved Mo and W-based electronic components and electrochemical sensors. To conclude, we address current challenges and future directions.

## 2. Transient Electronic Devices

### 2.1. The Arrival of Transient Technology

Triggered by the rapid technological innovation in microelectronics, one of the major challenges of the 21st century is managing and reducing e-waste [7]. According to the Global E-waste Statistics Partnership (GESP), a record amount of 53.6 million metric tons of e-waste were globally generated in 2019, from which only 17.4% reached appropriate management facilities. Electronic pollution, prompted by the buildup of hazardous e-waste, results in considerable accumulation of toxic substances (e.g., organic compounds and heavy metals) into the ecosystem and involves serious risks to human health [8]. Although the implementation of the Basel Convention (1992) has largely reduced the transfer of e-waste from developed countries to developing countries for final disposal, the exportation of second-hand electronic equipment for reuse and recycling has intensified [9]. Yet, during recycling operations, merely part of the equipment is reclaimed, whilst unrecoverable components and materials are all the same openly burned or dumped [10]. One of the most well-known e-waste dismantling and recycling centers is located in Guiyu (China) [11]. Guiyu is a leading example of how improper recycling methods can lead to severe environmental contamination, and exposure of workers and local residents to toxic substances [12]. One of the main culprits is the recycling of printed-circuit boards, profitable due to the prospect of retrieving integrated-circuit (IC) chips and valuable metals (e.g., gold, palladium, copper). Such activity implicates the disposal of molten lead and the release of extremely toxic wastewater [9]. Research on the broad-spectrum impact of scarce and inadequate e-waste recycling is ongoing.

The up-and-coming field of transient technology has quickly evolved in the past decade as a strategy towards realizing eco-friendly “zero-waste” electronic devices [3]. Device transiency allows electronic materials, components, and circuitry to undergo full or partial physical dissolution in a variety of different milieus [3]. Thus, risks and costs associated with e-waste disposal and recycling are minimized [2]. Unlike conventional electronic devices, these are designed to operate over a typically short and well-defined time frame before losing their functionality [2]. Initial research began with the coupling of conventional electronic components into degradable substrates, followed by the integration of partially soluble, and more recently fully soluble, electronic components into larger systems [13]. It was not until 2012 that Hwang et al. reported the first proof-of-concept transient electronic device composed of a thin silk substrate with overlaying Mg for the conductors, magnesium oxide (MgO) and silicon dioxide (SiO_2_) for the dielectrics, and silicon nanomembranes (Si-NMs) for the semiconductors [3]. The device included typical complementary metal oxide semiconductor (CMOS) components such as transistors, diodes, inductors, capacitors, resistors, interconnects and interlayer dielectrics. All parts dissolved in de-ionized (DI) water within 10 min. The same device, then sealed in silk packages, largely dissolved following subcutaneous implantation in Sprague–Dawley rats within 3 weeks. One of the device components, the metal oxide semiconductor field-effect transistor (MOSFET), later encapsulated within MgO layers and crystallized silk, showed stable operation for ≈90 h and a swift dissolution in DI water thereafter.

Successful developments to the ground-breaking work of Hwang et al. have launched important routes towards establishing electronic device transiency in a variety of compelling applications. Physical transiency is an asset, not only in the framework of consumer electronics, but also in the circumstance of temporary environmental monitors and medical implants [14]. Soluble environmental sensors could enable the close monitoring of ecological factors or pollutants, without further contributing to the release of toxic dissolution products into the ecosystem [15,16]. These devices could have a tremendous influence in the future of ecological conservation and agriculture. With respect to consumer electronics, biodegradable eco-friendly devices provide an elegant solution to alleviate e-waste and landfill [17,18]. As emphasized in the following section, the pertinence of device transiency is well portrayed by implantable biomedical devices and systems [14].

### 2.2. Transient Bioelectronic Devices

Biomedical devices seek to increase patient quality of life, expand longevity, assist natural body function and monitor physiological parameters [19]. Nowadays, they have become an integral part of common medical practice. Concurrent with increased life expectancy and technological innovation, there is a fueled demand for increased medical device functionality such as sensing, closed-loop health monitoring, programmable therapy, personalized medicine, and remote health monitoring [20].

Wearable devices (e.g., adhesive patches, smart wristwatches or chest straps) allow for short-term continuous ambulatory monitoring of human vital signs, which are of particular importance in the triaging and follow-up of certain medical conditions [19,21,22]. Nonetheless, wearable devices are in indirect contact with biological tissue, which heavily limits their therapeutic effectiveness [21]. Implantable devices, on the other hand, are fixed near the target organs and are mainly intended for long-term use. Due to long-term exposure, permanent implants may act as a focal point for infection (which may arise as a result of biofilm production and bacterial migration within the body [6]) and fibrotic capsule formation surrounding the implant. Therefore, permanent implants often require surgical extraction or replacement in the long run, exposing the patient to further complications and general discomfort [22]. Short-term transient implants, functional for days or weeks, as opposed to months or years as is distinctive of permanent implants, can circumvent these drawbacks.

Contrary to structural implants commonly employed as physical support structures such as bone screws, stents, fixation devices, or hip implants, those with integrated electronics are capable of diagnosis and treatment of various clinical issues [20]. By means of transducing physiological inputs into electrical outputs, these devices are capable of sensing, processing, communication and actuation [23]. By resorting to electronic implants with sensing capabilities, physio-pathological data of therapeutic interest could be locally acquired and transmitted to an external receiver, in a clinical setting, or remotely [24]. Physically-transient equivalents may serve as temporary diagnostic tools in the eventuality of clinical injuries that are also transient in nature (e.g., post-surgery monitoring, wound healing, tissue regeneration) [25] or as vital parameter monitors [6]. Within the spectrum of sensing mechanisms, electrical and electrochemical sensors play an important role as diagnostic tools in point-of-care testing owing to their operational simplicity, low cost, ease in miniaturization, high sensitivity, and specific sensing with low limits of detection [26,27]. Notorious diagnostic tools based on electrical and electrochemical principles include blood glucose monitors, neurotechnological implants, and cochlear implants [28]. Considering a post-operative status, transient electrochemical sensors could enable the real-time continuous monitoring of biochemicals, allowing medical personnel to swiftly respond if the need arises. Once the critical therapeutic window has passed, the transient device would steadily lose its functionality and dissolve in the body, avoiding complications and costs related to permanent implants and surgical removal.

Remarkable examples of transient electronic implants with sensing capabilities include a collection of transient sensors (not merely electrochemical) capable of cardiovascular monitoring [29], real-time monitoring of nitric oxide [30], temperature and pressure signal recording in the brain [31] and tumor hypoxia detection [32]. Unfortunately, these devices remain bound to limited in vivo device lifetimes, between minutes and days, as it is still a challenge to concurrently satisfy both sensing performance and gradual physiological dissolution [30].

#### Design and Fabrication Requirements

Serving as medical implants, transient electronics devices must meet certain criteria in terms of design, fabrication, and material selection. (1) Implantation, conceded via device miniaturization, grants a direct contact with the target tissue, and hence: in-situ signal acquisition, higher bio-sensing accuracy, and improved therapeutic efficacy. (2) The device should be biocompatible, i.e., should be able to properly function in vivo without eliciting adverse immunological responses, immunogenicity, or cytotoxicity [33]. Thus, prior to considering a material to be clinically viable, certain standards must be met in terms of biosafety, cytotoxicity, inflammatory response, and biofouling [33]. Specific to transient devices, these must also be bioresorbable. Following material degradation in vivo, all dissolution products should be capable of being resorbed, phagocytosed, or metabolized by the body [20,34]. Bioresorbability guarantees the absence of material bio-accumulation, which in turn eliminates the need for secondary surgeries intended for device removal [5]. (3) The biomedical device should be conformal and capable of withstanding the dynamics of physiological environments [6]. Bioresorbable implants fabricated on flexible substrates (e.g., polymers) [25] are capable of adjusting to the curvilinear shape of biological tissue and of minimal tissue irritation. Moreover, they are capable of avoiding tissue obstructions caused by conventional rigid device designs and limited sensitivity from poor device tissue contact [23]. As such, monitoring of human vital signs with high bio-sensing accuracy and potential therapeutic efficacy becomes achievable, and with it: patient self-monitoring, remote patient observation, relief of healthcare systems, fewer hospital re-admissions, and fewer medical waste. (4) Manufacturing techniques must be compatible with the use of transient materials:a substrate [6] responsible for the mechanical support of the device, material integration and handling, andelectronic (active and passive) components, employed in the thin film format to assure flexibility and sensible degradation time frames [25].

Biodegradable polymers give way to “soft” and flexible substrates [25]. Biodegradable polymers are often utilized in healthcare as resorbable sutures, tissue engineering scaffolds, stents and patches [34]. Some of the most investigated hydrolytically degradable polymers are poly-lactic acid (PLA), poly-glycolic acid (PGA) and poly-lactic-co-glycolic acid (PLGA) [6]. PLGA holds an excellent biocompatibility and releases toxicologically safe by-products as a result of bio-dissolution [35]. The fabrication process of PLGA is highly versatile, which facilitates the integration of electronic components [35]. A particular advantage of the PLGA co-polymer arises from the possibility to tune certain properties such as molecular weight, hydrophobicity, and co-polymer composition (ratio of lactide to glycolide) to diminish water adsorption and consequently, reduce degradation rates.

Published studies on the in vitro degradation rates of thin (10 µm) PLGA films (co-polymer ratio of 75:25) over the course of 10 weeks in phosphate buffer solutions (PBS) at 37 °C showed a steady mass loss (11%) after 6 weeks and approximately 35% mass loss after 10 weeks [36]. A higher content of hydrophilic glycolic units (co-polymer ratio of 50:50) facilitated the absorption and diffusion of water, and therefore, led to increased degradation rates [36]. An additional actor in the degradation profile of PLGA is pH. With PLGA biodegradation, the release of free lactic and glycolic acids tends to decrease the pH of the surrounding medium. This reduction in pH (≈ 7 to 2.4) sharply accelerates the PLGA degradation process [37]. However, the functional lifetime of a transient electronic device is defined by the individual and joint degradation rates of both the substrate and locally integrated electronic (metallic) components. Optimal rates vary according to the application in hand. In a later work of Hwang et al., in 2014, the integration of a transient hydration sensor onto a PLGA (15–20 μm thick) substrate was successfully achieved [38]. Whilst the electrodes and interconnects forming the sensor largely dissolved during immersion in PBS (pH 7.4, 37 °C) within the first 2 days, the PLGA substrate was estimated to only fully dissolve over the course of several months. The decrease in pH of the surrounding medium, as a result of the polymers’ acidic dissolution products, might influence the degradation rates of the superimposing metals. For instance, iron (Fe) is a biodegradable metal that natively degrades more quickly in acidic solutions (pH 5), rather than in more basic solutions (pH 7.4). Thus, if Fe thin films were to be deposited on top of PLGA, as part of a transient electronic device, the degradation of PLGA might trigger a faster Fe dissolution than anticipated. Alternatively, silk and cellulose derivates may likewise serve as biodegradable substrates. Silk offers flexibility, transparency, robust mechanical properties, the possibility of tailored dissolution rates and non-inflammatory amino acid by-products usable in cell metabolic functions [20]. Kim et al. reported the combination of silicon electronics (based on Si-NMs) and biodegradable thin substrates of silk protein [39]. Implantation in mice showed partial dissolution in 2 weeks, as well as a lack of inflammation around the implant site. Water soluble cellulose derivatives are mostly exploited in the context of printed and paper-based electronics [17]. Hwang et al. has successfully developed a transient CMOS system on rice paper (≈ 200 μm thick). Full dissolution occurred following immersion in DI water at 37 °C for 2 days [38].

Apart from the substrate, electronic components are compulsory for the functionality of any active device [6]. In conventional microfabricated systems, thin films of inorganic materials are typically employed as conductors, semiconductors, and dielectrics [6]. As attractive conductor elements, metals benefit from high electrical conductivity and stability [4]. Interest in biodegradable metals has recently shifted from simple biomedical structural components to thin film conductors, electrodes, interconnects, and interlayers, to be implemented in transient devices with active functionality [34]. Electronic components employed in the thin metal format should guarantee flexibility, appropriate dissolution time frames, and minimal physiological release of metal ions. Essential for their commercialization, thin metal films should be able to mechanically deform on top of malleable substrates without compromising their electrical and mechanical properties. Biodegradable conductors include alkaline metals, transition metals, and their alloys [20]. Paired with semiconductors and dielectrics, thin metal films of Mg, Zn, Fe, Mo, and W can form essential electronic components for CMOS circuits [34]. Nonetheless, Mg and Zn are known for dissolving very rapidly (≈ 0.5–3 μm h^−1^ [4]) in biofluids [20]. A too quick degradation and loss of mechanical integrity is a heavily restricting feature in functional implants [6]. Mo and W exhibit comparatively slower dissolution rates. Recent studies [4] have attested the foreseeable and controlled dissolution of Mo and W in biofluids at physiological temperature (<0.02 μm h^−1^). A controlled dissolution profile is key in ensuring stable, or at least predictable, signal outputs during device degradation. As with most metals, Mo and W benefit from a series of versatile fabrication methods, compatible with industrial-scale microfabrication techniques. Some of the most widespread fabrication methods for attaining Mo and W metal layers/nanostructures are: chemical vapor deposition (CVD) [40], physical vapor deposition (PVD), e.g., magnetron sputtering [41], wet chemistry [42], and hydrothermal methods [43] etc.

Encapsulation layers (hydrophobic polymers or inorganic materials), as packages or more complex permselective membranes, typically serve to shield soluble metallic components from water infiltration, halt physiological corrosion and dissolution, extend device life-times, minimize biofouling effects, and increase molecular selectivity [24]. Given that in electrochemical sensors the target analytes must be in direct contact with the electrodes responsible for molecular sensing, these cannot be encapsulated [24]. Thus, to assure a stable in vivo operation and appropriate device life-times, there is an urgent need for bioresorbable metals that can withstand direct contact with biofluids for longer periods of time, such as Mo and W.

Contrary to well-established microelectromechanical systems (MEMS) fabrication technologies, the attainment of transient electronic components requires some adjustments in terms of manufacturing strategies [14]. Whilst conventional electronics benefit from unlimited device designs, transient electronic systems are bound to the pre-requisite of biodegradability, which heavily narrows material selection [20]. The choice of adequate device materials should be based on their ability to physically dissolve upon an external trigger or stimulus: aqueous solution(s), light, temperature, pH, mechanical force [2]. Common microfabrication processes (deposition, patterning, doping, etching) are typically performed under harsh conditions, which are intrinsically incompatible with dissolvable materials highly sensitive to temperature, air, water, or corrosive chemicals [14]. To circumvent these limitations, a notorious strategy for the assembly of biodegradable electronic devices is to separately synthesize the biodegradable substrate and fabricate the electronic components, a technique known as transfer printing [44]. Here, layers of thin metal films may be obtained resorting to conventional fabrication techniques directly on a temporary Si substrate, between two polymeric sacrificial layers. Following dissolution of the sacrificial layers, the electronic device is released, “picked up”, and then transferred onto the final biodegradable substrate [44].

Despite recent advances, fully-bioresorbable electronic medical systems are yet to be applied in clinical settings or to become commercially available [45]. Unlike structural implants, it is expected that electronic devices are capable of communication with external receivers, for the continuous real-time monitoring of therapeutically relevant parameters [46]. The incorporation of active elements, however, requires a multitude of electronic components in addition to the sensing interface, including bioresorbable circuitry, wireless communication units, wires, and a power source [6]. Published, active implantable systems remain partly transient, predominantly due to the presence of non-bioresorbable electronic components: externally mounted via transcutaneous wires or meant for later surgical removal. In short supply, fully transient electronic components suitable for incorporation in larger medical systems can be found in the literature [30]. For instance, one can find examples of miniaturized transient batteries meant for self-sustaining transient implants [18]. Alternative powering methods (e.g., inductive coupling, piezoelectricity, electromagnetic) have been suggested too [46]. Nevertheless, these approaches suffer from limited implantable sites, biocompatibility issues, complex fabrication processes, biological side effects [46] etc. As emphasized in this review, bringing together existing bioresorbable electronic components and sensing interfaces (as illustrated in Figure 1) could lead to novel, self-sustained, fully-transient electronic implantable systems. Existing key sub-platforms will be discussed further. Given the advantageous properties of Mo and W, which will be discussed in detail further on, these bio-metals have been increasingly in the focus of scientific attention.

## 3. Bioresorbable Metals Mo and W

### 3.1. Biocompatibility and Bioresorbability

Bio-metals (i.e., biocompatible metals) are a class of biomaterials of particular interest for the design of electronically-active implants, where considerable attention is given to in vivo resistance towards corrosion and wear [47]. Due to their advantageous mechanical properties [34], bio-metals are regularly employed as load-bearing or support structures [5]. The corrosion of metallic biomaterials is given by material deterioration and loss when exposed to the hostile electrolytic environment of body fluids, giving rise to, e.g., metal cations, oxides, hydroxides, and phosphates [48]. When implanted, the corrosion of bio-metals is speculated to have a large influence on the release of metal ions and debris, which may have toxic effects on biological tissue [49]. The formation of a surface oxide passivation layer hinders the release of metallic ions and further corrosion of the underlying metal [48]. Nonetheless, physiologically present inorganic ions such as chlorides are highly effective in breaching the oxide passivating layer. The excess release of metallic corrosion products, above certain tolerance levels particular to each metal, may disturb the physiological homeostasis at the site of implantation [47], potentially resulting in metal poisoning and inflammatory reactions [33].

The corrosion of bioresorbable bio-metals requires further considerations. Bioresorbable bio-metals are expected to corrode in a controlled manner, accompanied by the release of metabolizable, innocuous, dissolution products. In order to prevent a hasty release of metallic ions and their bio-accumulation, soluble bio-metals should exhibit appropriate in vivo degradation rates and mechanisms [33]. The bio-accumulation of metallic corrosion products is highly dictated by their ability to dissolve in biofluids. Certain metal salts, however, exhibit poor solubility in aqueous environments and their biological elimination may be challenging [47]. Hence, it is important to systematically investigate the biological impact of these metals prior to their implementation in transient electronic implants.

Mg, Zn, and Fe are physiologically occurring bio-metals, largely considered in biomedical frameworks [50]. Mg biocompatibility and bioresorbability has been extensively studied and is well-established. Fitting with traditional microfabrication techniques, commercially available Mg-based bioresorbable implants include a stent system, bone pins and screws [5]. Nonetheless, Mg alloys tend to corrode non-uniformly and are prone to cracking due to stress corrosion [5]. Zn and Fe have not been as extensively used in transient implants, as research on their bioresorbability is still in progress [5]. Fe in vivo dissolution products are mainly insoluble: resulting in very sluggish physiological degradation rates, which might lead to long-term rather than temporary platforms as sought after, and likelihood of device material bio-accumulation [5]. Although Zn has excellent biocompatibility, it exhibits poor mechanical strength, stability, and ductility [5]. Alternatively, Mo and W dissolve steadily in biofluids, resulting in thin soluble corrosion products [51]. Due to their mechanical, electrical, and biocompatible properties, Mo and W have been appointed as promising candidates to form bioresorbable electronic implants.

Mo and W are unique transition metals located in the group 6B of the periodic table of elements [33]. Mo is a silvered-tone heavy metal (10.2 g/cm^3^ [33]) with high modulus, high thermal stability, and low electrical resistivity [41]. Thus far, Mo has been mainly explored in metallurgical applications. Mo is a trace element with distinctive chemical versatility and high bio-availability [52], attributable to the solubility of molybdate salts in water. Essential to a variety of microorganisms, plants, and animals: Mo occurs in the active sites of a wide range of metalloenzymes [53]. Moreover, found to have a vital role in human cell metabolism are physiologically-occurring Mo-containing enzymes [54]. Likewise, W is a heavy metal, with a density of 19.2 g/cm^3^, good conductor, flexible, and with a high melting point [55]. W has been mainly ascribed to industrial purposes and goods. Although less bioavailable and essential only to a limited number of bacteria: W is equally present in the active sites of metalloenzymes [56]. W is not a biologically-present metal. Nonetheless, human exposure can occur through various routes: occupation (mines and industries), medical equipment and devices (wires), environment (soil and water), and industrial goods (e.g., light bulb filaments, electronic devices, and ammunition) [55]. Mo and W-containing enzymes play important roles in the metabolism of nitrogen, sulfur, and carbon compounds [56]. As trace elements, both too high and too low amounts of these bio-metals in the body are detrimental [56]. Overall, the biocompatibility of W is far less understood.

Considering metallic implants, resorbability is dictated by the metabolic pathways of the constituent metals [51]. Dissociated from metal salts, metal ions may result in insoluble precipitates, difficult to extract from the body [49]. At physiological pH, the main leachable species of solid Mo and W are their bioavailable and metabolizable forms: molybdate anion (MoO_4_^2−^) and tungstate anion (WO_4_^2−^) [51]. MoO_4_^2−^ and WO_4_^2−^ are subsequently converted into metalloenzyme cofactors [45]. Mo is rapidly regulated and cleared from the body [5]. Excess Mo is excreted via urine 10–16 μg/L [51]. Whilst W is also rapidly cleared from the body via urine, within 24h after exposure, there is a tendency for W bio-accumulation in certain organs. Studies suggest that the breakdown of W-based medical devices result in elevated W serum levels and accumulation in bone [55].

In a published study conducted by Yamamoto et al. (1997), the cytotoxicity of Mo and W chlorides (metal salts) were investigated against murine fibroblast and osteoblastic cells [49]. The half maximal inhibitory concentration (IC_50_) values for the beforementioned cell lines were defined as ≈ 119 mg/L and 62 mg/L. Mo can, therefore, be considered in bone implants. A recent investigation conducted by Redlich et al., (2021) aimed to investigate the biocompatibility of Mo, focusing on stent applications [5]. Here, in vitro apoptosis and necrosis assays for human coronary artery endothelial cells and human coronary artery smooth muscle cells were conducted via a concentration series of Mo trioxide extracts (MoO_3_) in cell culture medium. Results showed that Mo ion concentrations, within values expected for a dissolving pure Mo stent (20 µm/year [5]) did not trigger apoptosis, necrosis, cytokine expression or thrombocyte activation. Mo was also colonizable by these cells, which were able to attach as quickly as 6h after incubation [5]. Overall, the toxicity of Mo is negligible [33]. Mo possesses good resistance to corrosion, antibacterial properties, and excellent mechanical and biological biocompatibility. It is, therefore, considered suitable for clinical applications [33].

Published studies by Peuster et al. (2003) revealed the absence of any cytotoxic effects following W coil implantation, despite elevated W serum levels [57]. Primary cultures of human pulmonary arterial endothelial cells, smooth muscle cells, and human dermal fibroblasts were incubated alongside ascending concentrations of W in cell culture medium. It took ≈ 25 × 10^4^ times higher concentrations of W, with respect to normal serum values (0.0002 mg/mL), to observe any cytopathological effects on the above-mentioned cell lines [57]. The authors further speculated that given the observed W coil degradation rates (29 µg/day), it would be highly unlikely that physiological W levels would ever reach a toxic threshold [57]. Although further research is required in terms of the long-term effects of W exposure, these studies reinforce the prospective use of W in transient electronic implants.

With emerging transient devices, there has been an increasing body of research on the physiological dissolution rates of biodegradable metals. These values must be in the right order of magnitude to avoid metal bio-accumulation and possible toxicity. Hence, it is important to not only assess the biocompatibility of bioresorbable metals, but also their dissolution mechanisms and kinetics.

### 3.2. General Dissolution Mechanisms

Electrochemical studies have proven that distinct metals offer different resistance to corrosion, sparked by their ease to lose electrons and form ions in solution [47]. Given the major influence that corrosion resistance has on the degradation of bio-metals in biofluids, bio-dissolution studies of transient metals are ongoing [58]. Dissolution profiles intrinsic to each metal, dictated by dissolution mechanisms and kinetics in biofluids, might be more or less favorable according to a specific application [20]. In general terms, the dissolution mechanisms and kinetics of transient materials varies with temperature, pH, type of solutions (DI water or simulated biofluids), oxygenation levels, concentration of ions, etc. In the case of thin metal films, additional factors such as porosities or local defects may lead to different dissolution behaviors than the ones anticipated by their bulk counterparts [58].

A published study conducted by Yin L. et al. (2014) [4] on the transient behaviors of dissolvable thin metal films, provided a much-needed starting point towards assessing the utility of the abovementioned bio-metals in miniaturized water-soluble systems (Figure 2, Figure 3 and Figure 4). These studies focused on the electrical conductivity, thickness, morphology and surface chemistry of Mg, Mg alloy—AZ31B (3 wt% Al and 1 wt% Zn), Zn, Fe, W, and Mo thin films. The experimental work encompassed different milieus, such as DI water and simulated biofluids (Hanks’ solution), basic and acidic pH, at both room temperature (RT) and physiological temperature (37 °C). For demonstration purposes, n-channel Si MOSFETs were built using electrodes and contacts originating from these metals. Patterned serpentine traces of thin metal films, 40–300 nm in thickness, were deposited via conventional microfabrication methods and patterned via photolithography and lift-off on glass substrates.

The thin film dissolution behaviors are represented as a time-dependent change in resistance (Figure 2a–g). Electrical dissolution rates (EDR) were defined by converting changes in electrical resistance to an effective thickness and used to describe the transient behavior as a function of time (μm h^−1^). Mo exhibited EDRs roughly 3× higher at body temperature, in Hanks’ solution, and approximately 3× lower at neutral pH, when compared to acidic pH (Figure 2e). Mo oxides (MoOx) were formed as corrosion products. W thin film EDRs showed no distinction between DI water and Hanks’ solution, increased with physiological temperature, and decreased roughly 4× in acidic Hanks’ solution (Figure 2c). W oxides (WOx) were formed as dissolution products. Curiously, the EDRs of W were substantially impacted by the choice of metal deposition method. W thin films formed by CVD (Figure 2f) showed a significantly lower dissolution rate than thin films achieved via PVD (Figure 2c), in this case via magnetron sputtering. Overall, Mg, Mg alloy, and Zn thin metal films (Figure 2a,b,d) revealed much higher EDRs than those registered for the remaining bio-metals, which increased in Hanks’ solution, but not significantly with body temperature.

Variations in thin film thickness throughout dissolution in DI water appear in Figure 2h–i. Likely due to the formation of passivating surface oxides, Zn and Fe thin films exhibit an increase in thickness over time. In particular, the latter was unchanged over the course of one month observation (Figure 2i), hinting at the substantial retention of Fe oxide products. A completely different trend was registered in the case of Mg, Mg alloy, and Zn (Figure 2h). The thickness of these thin films quicky approached zero after a complete loss in electrical continuity [4]. Favorably, the thickness of W and Mo thin films rapidly decreased within the first couple of days after which, it stabilized and decelerated as a result of MoOx and WOx surface formation (Figure 2i). We can then infer the dissolution of thin metal films if given by a multi-step process: a faster initial dissolution of the underlying metal, which is then halted by surface oxide dissolution products, and the complete dissolution of residual oxides [4].

Across all bio-metals, dissolution rates were influenced by the formation of surface oxides. These residual oxide layers dissolve at a much slower pace in comparison to the underlying metal and can act as partially protective layers, halting their dissolution [4]. Overall, this study has attested the foreseeable and controlled dissolution of Mo and W in simulated bio-fluids. Mg and Zn dissolve relatively quickly in both DI water and simulated bio-fluids. The long-term retention of Fe oxide products renders this metal unsuitable for a large majority of transient electronic systems [4].

### 3.3. Surface Chemistry

Subsequently, the authors focused on investigating the surface morphologies of the thin metal films over the course of up to 31 days. The morphological evolution was assessed via scanning electron microscopy (SEM) and transmission electron microscopy (TEM) imaging, as well as by using X-ray photoelectron spectroscopy (XPS) analysis.

The surface morphology of Mo (Figure 3a–d) and sputter-deposited W (Figure 4a–d), in the course of dissolution in DI water, appears to be uniform with evidence of micropore formation up to 26 and 31 days, respectively, (Figure 3i and Figure 4i). XPS analysis (Figure 3j) revealed an initial MoO_3_ native oxide, followed by the presence of a mixture of valence oxides (Mo^4+^, Mo^5+^, and Mo^6+^). The same trend was reported for the case of W (Figure 4j). As time progressed, only Mo^5+^ and W^5+^ remained. Following the disappearance of the pure metal, the gradual dissolution of WOx and MoOx in DI water can be inferred by the slowly decreasing and continued presence of XPS signals for up to 70 days. Following this period, a residual 10–20 nm thickness and a dissolution rate of ≈ 0.2–0.5 nm d ^−1^ for MoOx and WOx, is estimated. Hydroxide (OH^−^) and water (H_2_O) are also observed on the thin film surfaces (Figure 3k and Figure 4k) [4]. Macroscopically, the dissolution of Mg, and Mg alloy, appeared to be uniform. During dissolution, however, microscopically there was evidence of micropore formation and an uneven surface. Due to the high solubility of Mg oxides and hydroxides (Mg dissolution products) in water, these completely dissolved in 1 to 4 days, translating into a rate of ≈ 5–8 nm d^−1^. Zn dissolved non-uniformly, with significant evidence of corrosion. Zn oxides lasted for 5 to 7 days, leading to a dissolution rate of 120–170 nm d ^−1^. As anticipated, Fe oxides did not show any dissolution for one month and resulted in a 10-fold thickness increase with respect to the initial thickness value.

In sum, the EDRs of W, Mo, and Fe thin metal films are much lower (≈ 0.0010–0.02 μm h^−1^) than those of Mg, Mg alloy, and Zn (≈ 0.5–3 μm h^−1^) [4]. These studies enabled a deeper understanding on the multiple biodegradable metals available, and consequently the selection of the best candidates considering degradation time requirements in different applications [4]. Naturally, too highly or too low dissolution rates in biofluids, limit the use of certain metals in bioresorbable electronic devices [6]. Gradual degradation rates, as seen in Mo and W, are preferred for implantation (controlled release of metal ions) and stable (electric signal) device functionality [4,51].

Later, in 2020, Redlich et al., studied the effect of different electrolytes in the electrochemical corrosion behavior of pure Mo (Figure 5) [51]. The intent was to verify the feasibility of utilizing pure Mo for structural implants, such as stents. The authors found the results to be in accordance with the previous experimental findings of Yin et al. Investigations on the corrosion properties of pure Mo were held in simulated physiological environments through immersion tests. Three cylindrical samples were submerged in different electrolytes: buffered and unbuffered simulated biofluids (SBF), 0.9% NaCl, and SBF + copper (Cu), at 37 °C for up to 100 days. The surface morphologies were observed by optical microscopy and SEM. A macroscopic change was visible after 20 days of immersion in buffered and unbuffered SBF (Figure 5a–d). SEM/Energy Dispersive X-ray Analysis (EDX) analysis of the sample surfaces verified differences in the composition of the layers. Samples immersed in SBF and SBF + Cu media revealed calcium phosphate and Mo oxide formation, whilst samples immersed in NaCl media only revealed oxide species formation, on the sample surface. The corrosion rates were found to be in the range of 5 μm year^−1^ in unbuffered and buffered SBF, 12.6 μm year^−1^ in SBF + Cu and 13.2 μm year^−1^ in buffered NaCl solution. The degradation rate discrepancies between SBF and NaCl media originate from the surface formation of calcium phosphate, possibly hindering the corrosion process. Addition of Cu in SBF media also resulted in increased dissolution rates, possibly due to polarization effects associated with Cu(II) reduction. In conclusion, bulk Mo showed a gradual and linear surface corrosion accompanied by the formation of thin soluble corrosion products. Thus, Mo is a promising bioresorbable metal that deserves to be considered in the context of biomedical devices.

## 4. Mo and W-Based Electrochemical Transient Devices

With the increasing evidence that Mo and W can successfully serve as biodegradable metals, their use in microfabricated devices has naturally grown as well. Electronic devices based on electrochemical working principles are particularly appealing for the fabrication of bioresorbable healthcare monitoring devices. Fundamental electrochemical components recurringly seen in active electronic devices are for instance batteries, wires, interconnects and electrodes for molecular sensing. All these components are indispensable to achieve a self-sufficient electronically active implant.

A fundamental requisite for the successful operation of implantable devices is an adequate and reliable power supply. Batteries are the standard powering tools of portable electronics and implants due to their high energy density, long shelf life and consistent miniaturization achieved in recent years [59]. Nowadays, a practically limiting aspect of transient devices is the need for external power sources that are mostly not biodegradable [23]. Additionally, mechanically flexible energy sources able to conform to the curvature of biological tissue offers further advantages. By resorting to soluble batteries or supercapacitors, self-powered implantable devices could be attained, enabling monitoring and sensing of therapeutically relevant analytes over appropriate periods of time, prior to full system dissolution [18]. Considering these implications, flexible, biodegradable, and biocompatible energy sources have been the target of increasing attention as they are capable of powering next-generation transient implantable systems [59]. Wires and interconnects play an important role in describing the electrical behavior of electrochemical systems. These components perform the essential function of allowing current flow between the different sections, ensuring the proper performance of the device. Electrochemical sensors are defined as devices capable of converting an input of physical, chemical, or biological nature into a measurable electrical output signals [60]. They are one of the most powerful tools for monitoring target analytes of therapeutic interest in the body. In this section, we will describe recent impactful bioresorbable electrochemical components based on W or Mo.

### 4.1. Bioresorbable Energy Devices

Supercapacitors, also known as electrochemical capacitors, are high-power energy storage systems with an almost infinite cycle-life and rapid storage and release of energy [61,62]. Supercapacitors have higher specific power but lower specific energy than most batteries. Used in combination with batteries or as an individual energy supply, supercapacitors can cover several orders of magnitude in energy and power [61]. Lee et al. (2019) reported the fabrication of a fully biodegradable and stretchable wire supercapacitor via the use of a biodegradable electrode and electrolyte materials (Figure 6) [63]. Here, a water-soluble Mo wire served as a current collector, anodized Mo oxide film as an electrode, poly(1,8-octanediol-co-citrate) (POC) as an encapsulation polymer, and polyvinyl alcohol (PVA) as a biodegradable electrolyte (Figure 6a). The embedding of the wire supercapacitor inside the elastomeric polymer POC, endowed the system with its stretchability. The fabricated wire supercapacitor, without encapsulation, exhibited a capacitance retention of 82% after 5000 repetitive cycles of charge/discharge, mechanical stability under systematic deformation and stable electrochemical performance under repetitive bending, knotting, and coiling (Figure 6b). The serpentine-shaped wire supercapacitor without encapsulation was found to have fatigue at 50% strain, although it returned to its original length at 30% strain. The serpentine-shaped wire supercapacitor encapsulated with POC film maintained the capacitance for strains in the 10% to 50% range and 91% capacitance retention after 1000 repetitive stretching/releasing cycles with 30% strain (Figure 6c). A red LED was successfully operated using three serially connected wire supercapacitors with an operation voltage of 2.4 V. Accelerated dissolution tests in DI water were performed for all materials (65 °C or 90 °C). The estimated mass change rate of the Mo wire was roughly 0.035 mg d^−1^ with a mass retention of 88.5% after 15 days. The dissolution rate of the elastomeric polymer was approximately 5.3 mg d^−1^, with a mass retention of 65% after 5 days. Finally, the dissolution of the POC-encapsulated Mo wire supercapacitor was evaluated in DI water at physiological temperature (37 °C). The device operated in a stable manner for 11 days, after which it lost its functionality.

In the work of Tang et al. (2011), MoO_3_ nanoplates had been prepared as an anode material for aqueous supercapacitors with high energy density and good rate behavior [64]. Later, in 2018, Huang et al. developed a fully bioresorbable Mg–MoO_3_ battery system [18] (Figure 7). It consisted of all soluble materials, including Mg, Mo, MoO_3_, sodium alginate hydrogel, PLGA, and a polyanhydride encapsulation layer (Figure 7a). The battery was able to provide a stable output voltage, up to 1.6 V, sufficient to light a LED for 16 *h* in PBS solution. The battery exhibited desirable capacity (6.5 mAh cm^−2^) and prolonged lifetime up to 13 days. Dissolution studies were performed in PBS solution. Apart from Mo, which required 10 days to achieve full dissolution at elevated temperature (85 °C), the remaining materials dissolved in 9 days (Figure 7b). Biocompatibility was evaluated via typical cell viability assays. No cytotoxic effect was recorded after a 5-day incubation of PLGA samples (with or without MoO_3_) alongside L-929 mouse fibroblast cells. Subcutaneously insertion on Sprague–Dawley rats further demonstrated in vivo degradability at the full system level (Figure 7c). The authors commented that the system in hand could potentially be used to achieve self-powered therapeutic systems for distinct clinical applications such as tissue regeneration, pre- or post-surgery monitoring.

Yin et al. (2014) reported a fully biodegradable water-activated primary battery (Figure 8) [65]. Mg foils were chosen as the anode material and Mo, W and Fe foils as the cathode materials (Figure 8a). A biodegradable polymer (polyanhydride) was chosen for packaging (Figure 8b–c). Within the Mg–X batteries, Mg-Mo produced 2.4 mAh from 1 cm^2^ active area (50 µm thick Mg, 8 µm thick Mo). A multi-cell stacked configuration of four Mg-Mo single cells was created to illustrate scalability. A Mo paste obtained from a mixture of Mo powder and water-soluble sodium carboxymethyl cellulose glue, provided the electrical connections between the individual cells. The multi-cell Mg-Mo battery was able to power a conventional LED with a threshold voltage of ≈1.6 V. Dissolution studies in PBS at 37 °C showed a full dissolution of the polyanhydride encasing and partial dissolution of the Mg and Mo foils after 11 days (Figure 8d). Full dissolution was achieved through an additional 8 days at an increased temperature (85 °C). These results demonstrate the feasibility of miniaturized powering devices of potential practical importance for biodegradable electrochemical systems.

### 4.2. Bioresorbable Electrical Connectors

Kang et al. (2016) developed bioresorbable multifunctional silicon sensors for the brain that continuously monitor intracranial pressure and temperature, essential to the treatment of traumatic brain injury [31]. The construction involves a 30 μm membrane of PLGA sealed against a supporting substrate of 60–80 μm thick nanoporous Si or Mg foil. The sensing interface is then connected via biodegradable wires (Mo, 10 μm thick) and interconnects (sputtered Mo, 2 μm thick) to a miniaturized externally mounted potentiostat for wireless data transmission. Immunohistochemistry at the brain tissue level, 2, 4 and 8 weeks after implantation showed no cytotoxicity from the sensor or its dissolution by-products. Dissolution studies showed a full sensor dissolution after 30 h upon insertion into an aqueous buffer solution at RT. In vivo and in vitro experiments demonstrated precise real-time track of pressure, temperature, motion, flow and pH.

More recently, Li et al. (2021) developed a flexible and biodegradable electrochemical sensor capable of nitric oxide (NO) monitoring, which is typically associated with inflammation, neurovirulence and cancer progression (Figure 9) [30]. The device consisted of a bioresorbable copolymer substrate composed of poly (L-lactic acid) and poly (trimethylene carbonate) (PLLA-PTMC), gold (Au) nanomembrane electrodes, and a biocompatible poly(eugenol) film as the selective membrane. A biodegradable paste made from a mixture of PLLA-PTMC and Mo particles served as the electrical connector to the testing wires.

Real-time monitoring was achieved in vitro, at cellular and organ levels, and in vivo, in the joint cavity of a rabbit for a 5-day period with a wireless data transmission system. The wireless circuit was immobilized on the thighs of the rabbits through surgical tape and connected to the transcutaneously implanted NO sensor. NO detection was based on chronoamperometry, via a three-electrode configuration and the current responses at different concentrations were measured. Results exhibited a wide sensing range (0.01–100 μM), a low detection limit (3.97 nM), a fast response time and good anti-interference characteristics. The sensing device achieved full physical transiency after 15 weeks in PBS solution at 65 °C and proper performance was sustained for roughly 7 days. Following implantation, the device disappeared in 8 weeks. Biocompatibility of the sensor was investigated via coincubation with human aortic vascular smooth muscle cells for up to 5 days, with excellent results. The same was verified at the implantation site. As a future direction, the authors discuss the development of a fully implantable and transient control circuit that can be wirelessly powered with biodegradable energy sources [30]. For the time being, only the sensing platform was developed.

### 4.3. Bioresorbable Electrochemical Sensors

Electrochemical sensors are defined as devices capable of converting an input of chemical or biological nature into a measurable electrical output signals [60]. Relevant parameters include pH, serum metabolites, among others [66]. The field of biosensors was introduced in 1962, with the establishment of the glucose oxidase (GOx) enzyme-based electrochemical biosensor [67]. A biosensor is composed of a biological component (the analyte), a biological recognition element (enzymes, proteins, or receptors), a transducer (the electrode) and a signal processor. The transducer is responsible for producing the signal, in proportion to the analyte-bioreceptor interaction, and the signal processor for the collection, amplification and display of the signal [60]. These sensors offer high selectivity due to the biorecognition element [60]. GOx is an oxidoreductase enzyme that catalyzes oxidation/reduction reactions [66]. In (bio-)electrochemistry the oxidation or reduction of an electroactive species produces a measurable current (amperometry), charge accumulation or potential (potentiometry) or modifies the conductive properties of the solution in which the electrodes are inserted (conductometry) [68]. Electrochemical sensors are nowadays commonly used for clinical applications (e.g., diabetic and cardiac self-monitoring) and pharmaceutical research [60]. Examples of Mo and W-based electrochemical biosensors intended for metabolite detection are poor. The authors found only a couple of successful electrochemical sensors centered on Mo and W for physiological parameter monitoring.

Cordeiro et al. (2018) developed an enzyme-based glucose biosensor capable of in vivo brain monitoring, based on Au-coated W needle microelectrodes [69] (Figure 10). The W microelectrodes were coated with Au, a permselective Nafion membrane and a glucose oxidase (GOx) contained in a hydrogel (Figure 10a–c). W, Au and W-Au microelectrodes could detect small changes in H_2_O_2_ concentration, with a fast response time and high sensitivity.

Amperometric glucose biosensors were fabricated with different enzyme loadings (0.2, 0.4, and 0.6 U/µL) and evaluated against glucose concentrations up to 64 mM. The biosensors displayed a non-linear increase in oxidation current in response to glucose (Figure 10d). The biosensors were considered suitable for brain glucose detection in the physiological range (0.5 and 2.5 mM), with a limit of detection between 96.32 and 222.84 µM. The in vivo evaluation was performed following implantation in the medial prefrontal cortex of male rats. Brain glucose levels were modulated through local administration of glucose or by administrating insulin intravenously. In vivo and real-time changes in glucose levels were successfully monitored. However, the usage of non-biodegradable materials such as Au (as a constituent material of the microelectrodes) and Nafion (as a permselective protective membrane) prevent a full bioresorbability. In vivo solubility of the device, even if partial, was not accessed.

## 5. Conclusions

Transient technology is a thriving research topic with the ability to revolutionize modern-day electronic devices applicable in the environmental sector, healthcare, security, and consumer goods. Healthcare systems could highly benefit from the use of electronically-active implants, particularly those capable of monitoring physiological parameters, post-surgery, post-trauma, etc. These devices potentially circumvent secondary device removal surgeries, minimize risk of infections, reduce e-waste, and enable remote-health monitoring.

As the back-bone of transient devices, there has been an increasing number of systematic studies on the physiological biocompatibility, bioresorbability, corrosion, and dissolution profiles of bioresorbable metals. The most appropriate choice of metals is largely dictated by the final application of the device. Mg, Zn, and Fe are by far the most commonly employed bio-metals in transient (functional and structural) implantable systems, partly due to their well-established biocompatibility. Unfortunately, the near immediate physiological dissolution of Mg and Zn thin films, falls short of what is expected for transient electronic devices that aim to function for up to several weeks. In contrast, the sluggish dissolution profile of Fe and related insoluble dissolution products is equally limiting. Alternatively, a growing body of evidence has nominated Mo and W as extremely promising candidates to be considered in transient electronic devices.

Specific to electrical and electrochemical sensors, where active sites must operate in direct contact with biofluids, Mo and W offer significant advantages when compared to Mg, Zn, and Fe. Mo and W thin metal films: (1) show a gradual and predictable degradation in biofluids at 37 °C; (2) when in contact with biofluids, form a thin uniform oxide layer, prompting longer device lifetimes; (3) are capable of withstanding the electrolytic environment of the body for longer periods of time without the need for encapsulation layers; (4) degrade into soluble and metabolizable dissolution products; (5) are electronically stable throughout dissolution; (5) when in excess, appear to quickly clear from the body following toxic metal exposure. Whilst the biocompatibility of Mo is far more accepted by the scientific community, the same cannot be said for W. Although studies seem to suggest that the toxic threshold of W largely exceeds regular serum levels, further in-depth studies on the short and long-term biological effects of W exposure are required. In this respect, Mg and Zn have an upper hand and are more readily-available for incorporation in biomedical systems.

Increasing interest from industry and academia has led to a boost (although still modest) in published electronic components based on Mo and W. In this review, we have discussed a few cases of Mo and W-based power sources, wires, interconnects and sensors. With a strong clinical appeal, transient implants with sensing capabilities require a multitude of electronic components for their proper, self-sustained functioning. In practice, implantable sensors today only uphold the physiological dissolution of the sensing interface (where contact with biological tissue is compulsory), and essentially remain partly transient due to the integration of rigid and/or non-biodegradable electronic components (platform-integrated or externally mounted). Thus, contrary to biodegradable structural implants, fully-transient electronic implants, capable of physiological sensing, processing, communication and actuation, have not yet been achieved, let alone commercialized.

The applicability of transient technology in healthcare is still taking its first steps. Most research remains confined to in vitro and animal studies. Although examples of remarkable transient implantable sensors can be found in the literature, they suffer from recurring drawbacks [24]: limited in vivo device lifetimes (hours-days); unstable electrical output signals as a result of hasty metal oxidation, corrosion, and dissolution; limited sensitivity due to the need for encapsulation layers; difficulty to concurrently satisfy accurate physiological sensing and suitable device lifetimes; evident lack of full-system transiency.

Moreover, the design and fabrication of bioresorbable electronic devices is highly complex, more so at an implantable scale [22]. After physiological exposure, the overall degradation time of an electronically-active transient device is dictated by the flexible (typically polymeric) substrate, while the operational lifetime of the device is essentially dictated by the dissolution rates of the metallic films. For the attainment, and subsequent commercialization, of such systems further improvements must be made in terms of industrial-scale microfabrication methods and device designs. Transfer printing is a well-established multi-step technique that has notoriously separated (1) the synthesis of the biodegradable flexible substrate and (2) the metal deposition and patterning processes involved in the fabrication of the overlaying electronic components. Yet, the compatibility between soluble materials and conventional microfabrication processes can be further improved to simplify the fabrication process and facilitate more complex designs and architectures. For instance, room-temperature magnetron sputtering may be employed to directly deposit thin metal films on biodegradable polymeric substrates that require low-working temperatures. Further research should also be undertaken in the sense of bringing to a minimum the number of electronic components that require implantation along with the sensing interface. For instance, wireless technologies (powering and communication) are attractive solutions to circumvent the need for implantable batteries and transcutaneous wires breaching the skin [22]. However, careful consideration must be given to operating frequencies in order to minimize energy absorption by the body [22]. Once device lifetimes begin to lengthen, the open challenge of on-demand, material dissolution will increase and require more attention from the scientific community.

By investing in Mo and W-based electrochemical sensors capable of molecular recognition and pairing these with biodegradable components, important steps are taken in the direction of next-generation, fully transient active implants. These “discrete” monitors could provide immense relief to healthcare systems and groundbreaking technological momentum in the field of bioresorbable devices. First, further research on the biocompatibility and bioresorbability of Mo and, particularly, of W thin films, is compulsory.

## Figures and Tables

**Figure 1 sensors-22-03062-f001:**
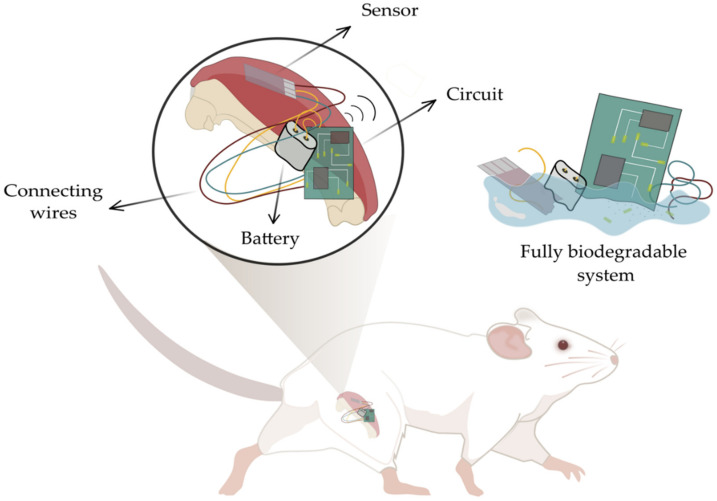
Illustration of a fully-transient electronically-active biomedical system in which all constituent parts (active sensing region, wires, interconnects, circuitry and power source) are biodegradable. Inspiration taken from BioRender.com (accessed on 25 February 2022).

**Figure 2 sensors-22-03062-f002:**
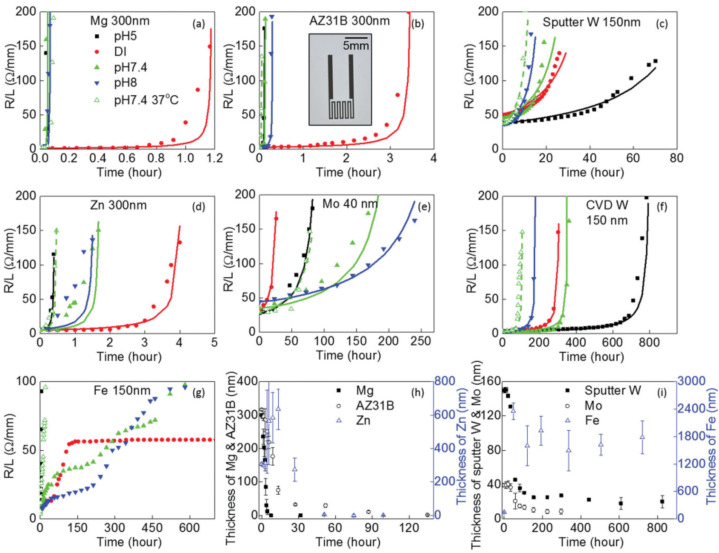
Changes in resistance (**a**–**g**) and thickness (**h**–**i**) of Mg, Mg alloy, Zn, Fe, Mo and W thin films as a function of time during dissolution in simulated biofluids or DI water, by varying pH and temperature. Reproduced with permission [4]. Copyright 2013, Wiley-VCH.

**Figure 3 sensors-22-03062-f003:**
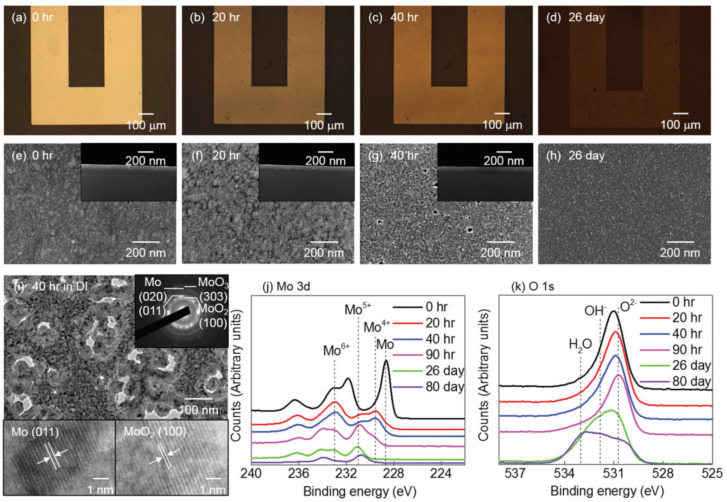
Evolution of microstructure and surface chemistry associated with dissolution of Mo in DI water. (**a**–**d**) optical images; (**e**–**h**) SEM images with cross-sectional views in the insets; (**i**) TEM bright field image with diffraction patterns and lattice fringes; (**j**–**k**) XPS data. Reproduced with permission [4]. Copyright 2013, Wiley-VCH.

**Figure 4 sensors-22-03062-f004:**
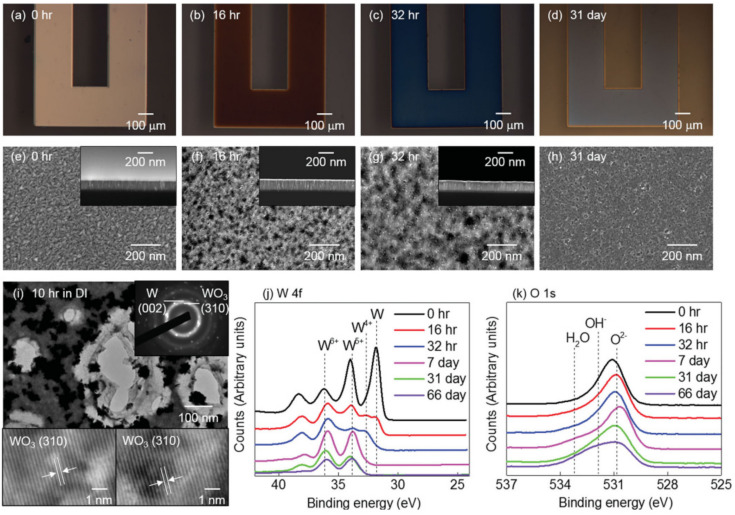
Evolution of microstructure and surface chemistry associated with dissolution of sputtered **W** in DI water. (**a**–**d**) optical images; (**e**–**h**) SEM images with cross-sectional views in the insets; (**i**) TEM bright field image with diffraction patterns and lattice fringes; (**j**–**k**) XPS data. Reproduced with permission [4]. Copyright 2013, Wiley-VCH.

**Figure 5 sensors-22-03062-f005:**
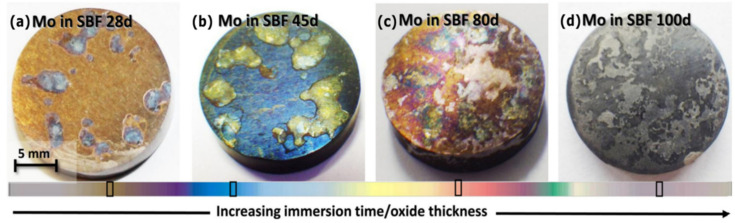
Macroscopical alteration of Mo samples immersed in buffered SBF for (**a**) 28, (**b**) 45, (**c**) 80, and (**d**) 100 days. Reproduced with permission [51]. Copyright 2020, Elsevier Ltd.

**Figure 6 sensors-22-03062-f006:**
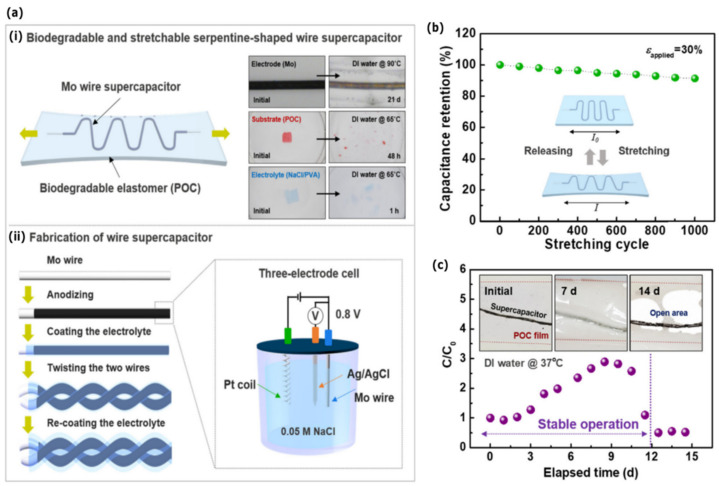
(**a**) Fully biodegradable and stretchable wire supercapacitor: (**ai**) Schematic illustration (left); Inset: optical images of the dissolution of constituent materials over time, in DI water (right); (**aii**) Fabrication process of the wire supercapacitor. (**b**) Capacitance retention under stretching by 30%. (**c**) Capacitance retention of the POC-encapsulated supercapacitor for up to 15 days. Inset: optical image showing the gradual disappearance of the encapsulation film. Reproduced with permission [63]. Copyright 2019, Elsevier B.V.

**Figure 7 sensors-22-03062-f007:**
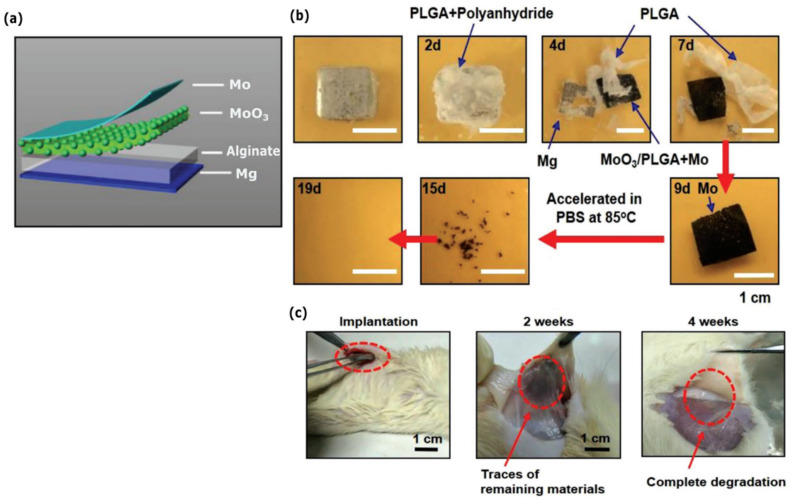
(**a**) Battery materials and configuration. (**b**) Optical images at various stages of dissolution of the battery in PBS. (**c**) In vivo dissolution studies in the subcutaneous area of SD rats for up to 4 weeks, suggesting full battery degradation. Reproduced with permission [18]. Copyright 2018, WILEY-VCH.

**Figure 8 sensors-22-03062-f008:**
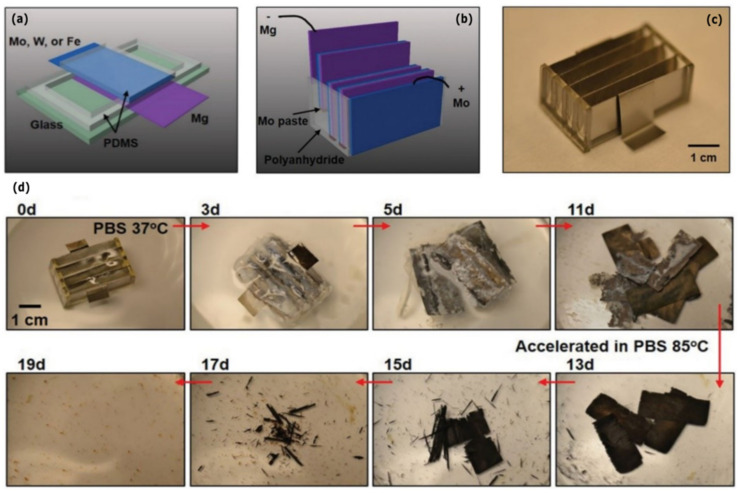
(**a**) Configuration of a single cell Mg-X battery for performance evaluation. (**b**) Configuration of a battery pack that consists of 4 Mg-Mo cells in series. (**c**) Optical images of the battery pack. (**d**) Optical images of the biodegradable battery dissolution behavior in PBS. Reproduced with permission [65]. Copyright 2014, WILEY-VCH.

**Figure 9 sensors-22-03062-f009:**
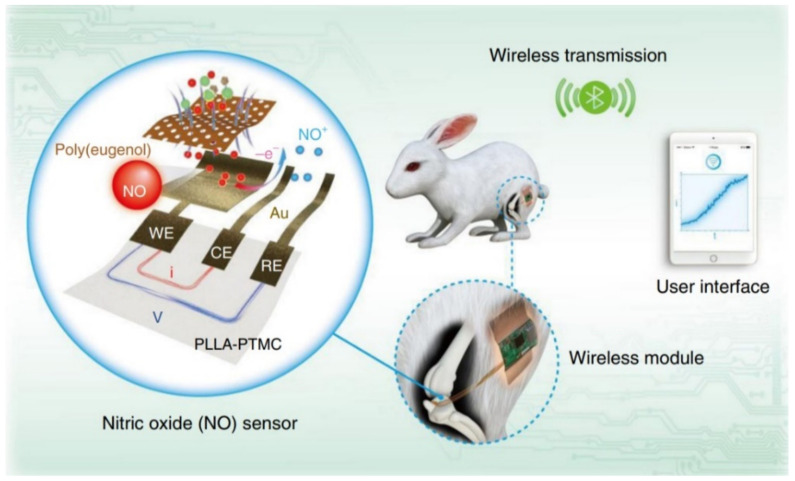
Schematic illustration of a transient NO sensor composed of a bioresorbable PLLA-PTMC substrate, Au nanomembrane electrodes, and a poly(eugenol) thin film. The implanted sensor is capable of continuously monitoring NO concentrations in vivo, as well as externally transmit the data through a wireless module [30].

**Figure 10 sensors-22-03062-f010:**
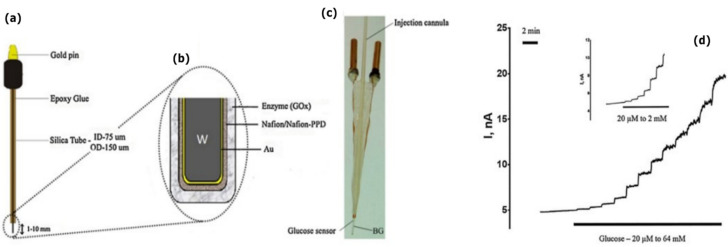
(**a**) W-Au needle type microelectrode. (**b**) Scheme of layer-by-layer assembly of the biosensor. (**c**) Implantable micro-biosensor device based on a W-Au based sensor. (**d**) Change in oxidation currents in response to the addition of low glucose levels (up to 2 mM). Reproduced with permission [69]. Copyright 2018, Elsevier B.V.

## Data Availability

Not applicable.

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
