# Peer review of "Biodegradable Molybdenum (Mo) and Tungsten (W) Devices: One Step Closer towards Fully-Transient Biomedical Implants"

_sensors, 2022, doi:10.3390/s22083062_

Round 1

Author Response

Comment# 1: Figure 2, Figure 6 and Figure 10 need to replace with improved images (in terms of image resolutions).

R: Thank you for your comment. Together with the initial manuscript we had submitted higher resolution (300 dpi) versions of the images/figures present in the Review article. Nonetheless, these have now been incorporated in the newest version of the article.

Comment# 2: Please recheck the Figure 8, there are numbers appearing in the background of Figure.

R: Thank you for your comment. This has been rectified. 

Comment# 3: Please recheck the format of all the references according to journal’s guidelines.

R: Thank you for your comment. This has been rectified.  All references were altered to the citation style “Sensors” in Mendeley Reference Manager.

Comment# 4: Please keep the figure notations same for all the figures. The notation of Figure 5 looks different from other figures. 

R: Thank you for your comment. This has been rectified to match the remaining figure notations.

Comment# 5: In conclusion part, please add more detail in terms of future directions related to biodegradable Molybdenum (Mo) and Tungsten (W).

R: Thank you for your comment. We have expanded the section “Conclusions” to add some more in-depth discussions and considerations on current challenges, possible areas of improvement, and future directions.

Reviewer 2 Report

The manuscript by Fernandes et al. provides an overview on the developments of Mo and W metals for biomedical implants. Mo and W represent emerging dissolvable materials towards transient implantable devices. A topical review of this active area is certainly worth consideration for publication. However, the manuscript does not have a smooth logic flow and requires substantial modifications on the arrangement. Some key aspects of this area are also missing. My specific comments are listed in the following:

  1. The contents of some sections are partially overlapped. As an example, the discussions on Mo and W bioresorbable metals (Section 3.1) are quite similar to the following section (Section 4). In addition, Section 6 should be combined with Section 5.
  2. What are the pros and cons of Mo and W among bioresorbable metals? Some in-depth discussions are highly encouraged.
  3. The commonly employed fabrication/processing techniques for Mo and W metals are worth mentioning to expand the scope of this review.
  4. The authors may want to describe the current challenges and speculate the future developments in this area.

Author Response

Comment #1: However, the manuscript does not have a smooth logic flow and requires substantial modifications on the arrangement.

R: Thank you for your comment. The initial sections of the Review paper have been largely re-arranged to improve the readability of the text, generate a smoother logic flow, and prevent repeating text. One can also find more in-depth discussions within each section.

Comment #2: The contents of some sections are partially overlapped. As an example, the discussions on Mo and W bioresorbable metals (Section 3.1) are quite similar to the following section (Section 4). In addition, Section 6 should be combined with Section 5.

R: Thank you for your comment. This has been rectified.

Comment #3: What are the pros and cons of Mo and W among bioresorbable metals? Some in-depth discussions are highly encouraged. The commonly employed fabrication/processing techniques for Mo and W metals are worth mentioning to expand the scope of this review. The authors may want to describe the current challenges and speculate the future developments in this area.

R: Thank you for your comments and valuable suggestions. The scope of the Review paper has been expanded to include: examples of widespread fabrication methods for attaining Mo and W metal layers, considerations on how conventional MEMS fabrication technologies must adapt to soluble materials, and current fabrication techniques for bioresorbable electronic devices, emphasizing the transfer printing technique. Please refer to the re-submitted manuscript section “2.2.1 Design and fabrication requirements” for the in-text alterations (in red).

Moreover, the “Conclusions” section of the Review has been considerably expanded. We start by reminding the reader of the value of device transiency in the context of healthcare and, specifically, implantable electronic systems. Next, we emphasize the growing body of evidence which point towards Mo and W as promising metals to be considered in the fabrication of transient electronic components. Although we make clear that the choice of most appropriate metals is largely dictated by the final application, we expand on the advantages and disadvantages of these metals in contrast with Mg, Zn, and Fe. Lastly, we elaborate on areas of improvement and future research necessary prior to the commercialization of transient electronic implants, in terms of: microfabrication techniques, operational lifetimes, alternative powering and communication (wireless) etc.

Reviewer 3 Report

The manuscript presented for an opinion is an interesting review. It contains a lot of data, illustrated with a collection of photographs and exploratory drawings. The sources of these photographs and graphics are presented, and I understand also with the appropriate approvals from the relevant publishers.
I recommend the work for publication. Only a part of the Conclusions should be slightly expanded. I would see in the next manuscript version a more extensive opinion on the future of the sensors in question in the upcoming applications, as well as new data on attempts to use them commercially. I also miss a more critical approach to the fact that the sensors in question are, however, based on heavy metals - which means that the biocompatibility of systems containing these elements (probably in the form of salts) is not so obvious. I would see the explanations for these points.

Author Response

Comment# 1: Conclusions should be slightly expanded. Comment# 2: I would see in the next manuscript version a more extensive opinion on the future of the sensors in question in the upcoming applications, as well as new data on attempts to use them commercially.

R: Thank you for your comment. Throughout the new version of the manuscript, we have added examples of published transient electronic sensors and clarified that, as compared to transient structural implants, these have not been commercialized. We further provide examples of impactful electrochemical sensors, which are not transient in nature, but have been commercialized and are extensively used in the clinic. Please refer to section “2.2. Transient electronic biomedical devices and 2.2.1 Design and fabrication requirements” of the new manuscript version, for the in-text alterations (in red).

The “Conclusions” section of the Review has been considerably expanded. We start by reminding the reader of the value of device transiency in the context of healthcare and, specifically, implantable electronic systems. Next, we emphasize the growing body of evidence which point towards Mo and W as promising metals to be considered in the fabrication of transient electronic components. Lastly, we elaborate on areas of improvement and future research necessary prior to the commercialization of transient electronic implants, in terms of: microfabrication techniques, operational lifetimes, alternative powering and communication (wireless) etc.

Comment# 3: I also miss a more critical approach to the fact that the sensors in question are, however, based on heavy metals - which means that the biocompatibility of systems containing these elements (probably in the form of salts) is not so obvious. I would see the explanations for these points.

R: Thank you for your comment. In the new manuscript version section “3.1 Biocompatibility and bioresorbability” we have made clear that indeed Mo and W are heavy metals. We have expanded on the risks associated with the biodissolution of heavy metals, in particular the non-solubility and possible physiological precipitation of metal salts, which might result in bioaccumulation of toxic compounds. However, at physiological pH, the main leachable species of solid Mo and W are their bioavailable and metabolizable forms molybdate anion and tungstate anion. Molybdate and tungstate salts have good solubility in water. We have added a study conducted by Yamamoto et al., (1997), where the cytotoxicity of Mo and W salts was investigated. Published studies have attested the fast regulation and clearance of Mo (especially) and W from the body, after metal exposure. However, it appears that W tends to accumulate in biological tissue, in particular bone. Nonetheless, studies show that even elevated W serum levels do not result in any form of cytotoxicity. We also added that, as a non-biologically-present element, the biocompatibility of W is still questioned, and that further research needs to be done prior to utilizing this metal in biomedical implants.

Reviewer 4 Report

The problem considered at work is very important and undertaken for many years by scientists. The article deals with the growing need for new devices biomedical applications. The problem is current and very significant in light of the continuous increase in the new medical equipment, as i.e. biosensors.

The manuscript is very interesting and well written. However, there are several points that I would like to address:

  1. Introduction: please precise what means the term “short-lived” by giving approx.. time of using this kind of device.
  2. Transient electronics: please precise medium time of degradation both for polymers and biodegradable metals and define a general conclusion how those two  “times of degradation” for completely different materials cooperate with each other on influence on each other.

Author Response

Comment# 1: Introduction: please precise what means the term “short-lived” by giving approx. time of using this kind of device.

R: Thank you for your comment. We have now clarified the expected operational lifetime of transient “short-lived” electronic implants to be between days and weeks i.e., sufficiently long to track, for instance, a post-operative patient status and dissolve thereafter. With transient implants, we attempt to move away from the costs and risk related to permanent implants, which aim to serve for months-years. Please refer to the re-submitted manuscript section “Introduction” and “2.2. Transient electronic biomedical devices” for the in-text alterations (in red).

Comment# 2: Transient electronics: please precise medium time of degradation both for polymers and biodegradable metals and define a general conclusion how those two “times of degradation” for completely different materials cooperate with each other on influence on each other.

R: Thank you for raising this enquiry. Undoubtably, the overall degradation time of an implantable transient electronic device is dictated by the dissolution kinetics and mechanisms of both the soft polymeric substrate and the overlaying electronic components (thin metal films). We have added some considerations on this matter. By taking the example of poly-lactic-co-glycolic acid (PLGA), we have clarified that the biodissolution of PLGA (substrate) in physiological environments may lead to an accentuated local decrease in pH, due to the release of acidic dissolution products. In the eventuality that the overlaying electronic components are made from metals whose dissolution rates are impacted by pH, this might lead to slower or faster dissolving thin metal films. In the case of Fe thin metal films, a PLGA substrate could speed up the dissolution rates of the electronic components, leading to a faster dissolving device, and shorter operational lifetimes. Please refer to the re-submitted manuscript section “2.2.1 Design and fabrication requirements” for the in-text alterations (in red). In the section “Conclusion”, we have complemented that the overall degradation time of an electronically-active transient device is dictated by the flexible (typically polymeric) substrate, while the operational lifetime of the device is essentially dictated by the dissolution rates of the metallic films.

Round 2

Reviewer 2 Report

The authors have carefully addressed all my concerns. I recommend the manuscript be considered for publication in its current form.